# Implications of the Use of Silica as Active Filler in Passenger Car Tire Compounds on Their Recycling Options

**DOI:** 10.3390/ma12050725

**Published:** 2019-03-01

**Authors:** Johannes W. van Hoek, Geert Heideman, Jacques W. M. Noordermeer, Wilma K. Dierkes, Anke Blume

**Affiliations:** 1Elastomer Technology and Engineering (ETE), Department of Solids, Surfaces and Systems (MS3), University of Twente, 7522 NB Enschede, The Netherlands; J.W.vanHoek@utwente.nl (J.W.v.H.); J.W.M.Noordermeer@utwente.nl (J.W.M.N.); A.Blume@utwente.nl (A.B.); 2Professorship for Polymer Engineering, University of Applied Sciences Windesheim, 8017 CA Zwolle, The Netherlands; G.heideman@windesheim.nl

**Keywords:** de-vulcanization, tire, recycling, car tire, sustainable, silica

## Abstract

Tires are an important vehicle component, as car handling, safety and fuel economy depend for a major part on the tire composition and construction. As a consequence, tires are improved continuously. The most prominent improvement in the recent past was the use of a silica-silane filler system in passenger car tread compounds, instead of traditionally used carbon black. For recycling and re-use of end-of-life car tire rubber one of the most promising recycling methods is devulcanization: re-plasticizing the vulcanized rubber by selectively breaking the sulfur bridges between the polymer molecules. In the present paper, the influence of silica, which is present in the passenger car tires granulate, on both devulcanization and subsequent revulcanization, is investigated. In a step-wise approach it is shown that the presence of silica influences both devulcanization and revulcanization. The best tensile strength of the revulcanizate, using a carbon-black-based revulcanization formulation, was 5 MPa. This could be improved to 6.5 MPa by using 2.8 phr of 1,3-DiPhenylGuanidine (DPG) in the revulcanization formulation. After addition of a silanization step during revulcanization by adding 3.2 phr bis[3-(TriEthoxySilyl)Propyl] Tetrasulfide (TESPT), a silane, to the formulation, the tensile strength of the revulcanizate was further improved to 8 MPa. With these results it is shown that the silica in the granulate can be used to improve the revulcanization properties. To check the benefits of using pure tire tread material for the devulcanization and subsequent revulcanization, of both a carbon black and a silica-based virgin tread compound, it is shown that a tensile strength of the revulcanizate of 13 MPa can be reached. This shows the potential of devulcanized rubber when the various tire components are separated before whole car tire material is granulated as the beginning of the recycling.

## 1. Introduction

The performance of passenger car tires is continuously improved, supported by governmentally imposed labeling systems on relative performance in terms of safety and environmental aspects, noise and fuel economy. Due to the sheer amount of 10${}^{9}$ tires/year produced worldwide, waste passenger car tires pose environmental problems when dumped because of their intrinsic resistance against decomposition, or when catching fire in the open air because of the sooth and fumes produced. These problems are addressed by the industry as well as academia as shown by the increasing numbers of patents related to reusing the materials of end of life tires by reclaiming, pyrolysis and related processes: Figure 1. Replasticizing waste tire rubber, as done in reclaim processes, aims at reuse of the valuable material for new products. However, due to the high shearing forces and processing temperatures applied in the conventional recycling processes, the reclaimed rubber is not readily reusable for high quality end products as scission of the polymer chains is taking place.

Finetuning of the process conditions leads to a shift towards devulcanization. It was shown that damage of the polymer chains can be limited considerably by tailoring the process conditions [1,2]. For EPDM [3] it has already been proven that devulcanization is possible with acceptable rubber properties of the revulcanized material. For end of life passenger car tire rubber this is still under development, mainly because Styrene Butadiene Rubber (SBR), a main component of passenger car tires, is more sensitive and prone to side reactions under the reclaiming process conditions.

An additional challenge is the production of a clean car tire granulate. During grinding, the rubber is separated from remaining steel, fibres and contaminations. The preferred particle size of the Ground passenger car Tire Rubber (GTR) is determined by the degree of purity required for the final application, and the costs. Besides, the granulate is a mixture of different rubber compounds from all tire components. An important, relatively recent, change in passenger car tire tread compositions is the use of silica as reinforcing agent rather than the more common carbon black. This is because of the significant improvement of wet grip and rolling resistance of the tires, leading to improved safety and better fuel economy of vehicles. However, as earlier work has shown, devulcanization of silica-based rubber is more difficult than of its carbon black counterpart, as indicated by a Horikx-Verbruggen analysis [4,5] to determine the ratio of polymer to crosslink scission. This evaluation however has its shortcomings when applied to a blend of different compounds. Within the present study it was decided to use stress strain properties after revulcanization as a measure of the quality of the devulcanizate.

The devulcanization process applied in this study is a thermo-chemical-mechanical twin-screw extruder based, continuous process, upscaled from a small scale batch process, as described by Saiwari [5]. In the latter study, a low shear process at a low screw speed was applied. In literature, both high and low shear processes using twin-screw extruders, are mentioned. However, most studies are based on a feedstock with a high Natural Rubber (NR) content, like truck tires, or a mix of passenger car and truck tires, as NR is a main component of truck tires. Presence and content of silica is not mentioned [6,7,8,9,10].

In previous studies, the devulcanization agent DiPhenyl DiSulfide (DPDS) was used, but due to the smell of the final product and a reported scission of the polymer chains [11], an alternative had to be found: 2-2${}^{\prime }$-Di-Benzamido-Diphenyldisulfide (DBD). This devulcanization aid is used for most of the research described in this paper. As a result of the use of DBD, and corresponding optimization of the devulcanization process conditions, the physical properties of the devulcanizates changed from severely staining and smelling into a dry, sticky, coarse powder, that could be transformed into a coherent slab by milling. However, after mixing of the pure devulcanizate with the revulcanization system an intensive homogenization on the mill was necessary as the material did not mix well enough in a laboratory size Brabender internal mixer.

Because of a relatively low tensile strength of the first batch of revulcanized material, the devulcanization process was thoroughly improved. As the feedstock contained a considerable amount of silica, the revulcanization process and compound formulations were optimized for this.

To study the influence of a mixture of all kinds of compounds as can be found in passenger car tire granulate, both a silica-based and a carbon black-based tread compound were prepared and stress strain properties before and after devulcanization (and subsequent revulcanization) were measured. By microscopy of the fracture surfaces of the revulcanizates, the influence of silica on devulcanization are shown.

## 2. Results and Discussion

### 2.1. The Presence of Silica in GTR

The introduction of silica in passenger car tire treads started more than 20 years ago [12]; therefore, whole passenger car tire granulate of recent origin must contain a certain percentage of silica. By Thermo Gravimetric Analysis (TGA), Figure 2 and Table 1, the amount of silica in a compount can be determined. Commonly the amount of polymers is defined to be equivalent to the weight loss of the sample between 300 ${\;}^{\circ }C$ and the point of a sharp decrease in weight at about 450–550 ${\;}^{\circ }C$. In Figure 2, the amount of volatiles, to include mainly process oils, wax and other lightly evaporating components, ranges from 100 → 79 wt% = 21 wt%, the polymers from 79 → 44 wt% = 35 wt%, carbon black from 44 → 10 wt% = 34 wt% and silica + ash from 10 → 0 wt% = 10 wt%. By setting the polymer content to 100, the amount of volatiles, carbon black and silica + ash can be easily converted into phr’s (grams per hundred grams of polymer).

With additional analysis it was shown that the silica content varies around an average of 23 phr. By Fourier Transform Infrared Spectroscopy (FTIR) analysis it was shown that the remaining GTR-ash after TGA clearly contains a certain amount of silica: curve B in Figure 3.

By Scanning Electron Microscopy-Energy Dispersive X-ray analysis (SEM-EDX), the presence of silica particles up to 250 μm could be shown, see Figure 4. This can be explained by the use of more conventional types of silica in the original tire compounds, with a dispersion not as good as that of more recent quality types.

### 2.2. Preparation of Devulcanizates in the Twin Screw Extruder

Devulcanization of GTR was done in a continuous process with the twin-screw extruder. The formulations as given in Table 2 were used. A disulfide concentration of 30 mmol/100 g polymer, which is equivalent to 6.85 wt% DBD, was found by Saiwari [13] to be optimal. In another study of Saiwari [2,5], a concentration of 30 mmol/100 g GTR was used. For DBD this is equivalent to 3.9 wt%. Both concentrations are used in this study. A concentration of TDAE of 6.2 wt% relative to GTR was found by Saiwari [5] as the best performing one.

Verbruggen [4] has shown that the Horikx diagram could be used with reliable results for filled single rubbers like EPDM. According to the assumption that a devulcanizate with a high amount of debonded sulfur crosslinks and a low degradation of the polymers has the best properties for reuse, it is reasonable to expect that devulcanizate with the best devulcanization parameters would show an optimized tensile strength after revulcanization. Devulcanized car tire granulate, with its complex composition of different compounds, several polymers, active fillers and a mix of granulate sizes, however, shows a different behavior: the correlation between the degree of devulcanization and the strength properties is not necessarily given. Therefore, the tensile strength of the revulcanizate was chosen as optimization parameter in this study and some of the test parameters of the preceding study [5], like the concentration of TDAE, were tested again. Also, a concentration of 1 wt% TDTBP was used, a similar concentration as was used in the previous mentioned studies.

Because of the powdery consistency of the devulcanization agents, swelling before devulcanization was not necessary. Just before use, the GTR was manually mixed with TDAE, TDTBP and DBD until the mixture was homogeneous. Mixing time was 5–10 min at room temperature, depending on the amount of GTR. The functions of TDAE are multiple: it acts as a processing aid during devulcanization, as an additive to facilitate the diffusion of devulcanization agents into the GTR particles, and to prevent dusting while dosing powdery devulcanization aids.

The screw design of the extruder was based on a minimal shear concept: minimize the application of shear during the processing at high temperatures and perform shear at lower temperatures after the chemical devulcanization, see Figure 5b. The extruder feed section was configured with conveying elements with a short flight to build some pressure, and mixing and kneading element to mix the additives thoroughly with the GTR. The temperature of this section, 130 ${\;}^{\circ }C$, is just above the melting point of DBD, 120 ${\;}^{\circ }C$, to melt this additive but to not start devulcanization already before the mixture enters the devulcanization section. In this section only small variations in screw-flight were applied but no kneading or mixing elements. The blend of rubber granulate, oil, stabilizer and devulcanization aid was gravity fed into the entrance section via a funnel, but care was taken that the funnel was always filled with the mixture to be sure that trapped air could be driven out by a constant flow of nitrogen. The extruder was operated under nitrogen atmosphere to prevent oxidation of the devulcanizate, which was supplied through a dosage point midway of the extruder, just before the start of the compression section. To release the overpressure, a ventilation point was foreseen, situated just behind the mixing section. Although some mixing and kneading elements were used directly after the pressure section, which was configured with elements with a short flight to be able to supply the necessary discharge pressure, these elements did not add any improvement and will be left out when redesigning the screw setup. Also, nitrogen was supplied to the die, as shown in Figure 5a. The devulcanizate was cooled down to approx. 60 ${\;}^{\circ }C$ directly after the extruder with the cooling calendar. The devulcanizate was milled at 60 ${\;}^{\circ }C$ and a 0.1 mm gap between the rolls of the mill until a homogeneous slab was produced. Subsequently, the devulcanizate (DGTR) was stored at room temperature for at least 24 h before further processing.

### 2.3. Revulcanization of DGTR

Initially, the revulcanizates were prepared with formulation 1 of Table 3 and compounded as described in Table 4. Because of the use of a carbon black-based revulcanization formulation at this stage of the investigations, just the amount of filler was of importance, so no differentiation was made between carbon black and silica. Vulcanization was performed under pressure in the Wickert press for t${}_{90}$+2 min at 170 ${\;}^{\circ }C$. The formulation was derived from a common, carbon black-based tread compound. The best tensile strength obtained with this series of devulcanizates was 5 MPa.

### 2.4. A Closer Look into the Influence of Silica on De- and Revulcanization of Tire Compounds

#### 2.4.1. Devulcanization of Silica-Based Tire Compounds

In a previous, unpublished, study the influence of a silica reinforced vulcanization system on the performance of the devulcanization process was investigated by S.Saiwari [14] using the Horikx method, as developed by Verbruggen [4] for devulcanization. In this study a tread compound with formulation as in Table 5 and the compounding procedure in Table 6, was devulcanized using DPDS. The range obtained is marked in Figure 6 as position 1: silica-based tread compounds showed a low decrease in crosslink density of 0.2–0.4 and corresponding sol-fractions on the random main chain scission line. Similarly typical results for carbon black-based vulcanizates are marked in Figure 6 as position 2: they usually show a decrease in crosslink density of 0.6–0.8 and corresponding sol-fractions close to the crosslink scission line after devulcanization.

This indicated that the amount of polymer that could be released from the polymer network of a silica reinforced compound by devulcanization is substantially lower than for a carbon black reinforced rubber. Possible explanations are, see Figure 7:The physical bonds between carbon black and the polymers as opposed to the chemical bonds between silica, silane and the polymers, which result in a different kind of network;The nature of the chemical bonds (mono-, di- or polysulfidic) present between the silane coupling agents and the polymers;A combination of these two.

As the applied devulcanization process does not break mono-sulfidic crosslinks, and the chemical bonds between silane and polymers are presumed to be primarily monosulfidic, a higher percentage of these type of links are supposed to be the main reason. In addition, it was found during the experiments of the present study, that the degree of devulcanization of tire granulate was not in relation to the tensile properties found after revulcanization of this material.

This must be due to the complex composition of the devulcanized granulate based on the mix of all tire components. Also, it is mentioned that unsaturated elastomers in general and SBR in particular are sensitive to recombination during devulcanization, especially for temperatures above 200 ${\;}^{\circ }C$, as were used in previously mentioned studies [16,17]. For this reason, the tensile strength of the revulcanizates was used as first parameter for optimization instead of the decrease in crosslink density.

#### 2.4.2. Revulcanization of Silica-Based Compounds

For silica it is known that its surface is covered with silanol moieties, as depicted schematically in Figure 8a with the consequence that it is acidic of nature and has a negative influence on the vulcanization reaction. Furthermore, to create a chemical bond with the polymer, a coupling agent is needed, mostly a silane, see Figure 8b [18]. As depicted in this figure, there are still free silanol moieties left on the silica surface which hinder vulcanization due to their acidic nature. It is well known that in silica reinforcing technology DPG, a commonly used secondary accelerator in vulcanization [19], is required to shield these free remaining silanol moieties, with the advantage of neutralizing the acidic character of the silica and hence improve the vulcanization of a silica-based compound [20].

However, as the amount of silica in the GTR is considerable, as described before, the question may be raised if there are free silanol moieties left or regenerated after devulcanization. Therefore, DPG was added to the formulation to compensate for the acidity of the silica. The amount of DPG was set at 2 phr, similar to the concentration used in the silica-based model compound, Table 3 formulation 2, but later increased to 2.8 phr after optimization, formulation 3. At this stage of the investigations it was necessary to differentiate between the amount of carbon black and silica, and the composition of the GTR was updated after additional analyses. Also, for better repeatability of the tensile test results, an additional 3 phr of sulfur and TBBS had to be added to the formulation, as shown in formulation 3 of Table 3. This might be due to the high viscosity of the 100% revulcanizate compound in relation to the migration of sulfur in the compound during the revulcanization [21]. Whether this is still needed when using the devulcanizate in blends with other elastomers will be part of future research. With these optimizations, the best tensile strength value increased to 6.5 MPa.

As the improved results due to the addition of DPG show, a certain amount of free silanol moieties is still present on the silica surface. To make use of these for building an additional silica network, a silanization step before the revulcanization process with a coupling agent (e.g., TESPT) is again necessary to further improve the tensile properties. By this secondary silanization, as shown in Table 7, the tensile strength further improved to 8 MPa after vulcanization for t${}_{90}$+2 min at 170 ${\;}^{\circ }C$ under pressure in the Wickert press (see Figure 9).

These results demonstrate that the first improvement by the addition of DPG alone remedied a certain acidity due to the silica in the GTR and hence in the devulcanizate. The second improvement by the silanization with additional TESPT indicates that the silica still has free silanol moieties left for a second silanization, enabling a further improvement of the properties of the material.

### 2.5. Benefits of Using Pure Tread Compounds, Carbon Black- as well as Silica-Based

To emphasize the benefits of separating the different tire components and using pure tread compounds, an additional study into these was done. As was discussed before, silica reinforced compounds are more difficult to be devulcanized than carbon black reinforced ones. In the earlier study [14] the material was not analyzed in terms of stress-strain properties. To study the effect of silica as reinforcing filler compared to carbon black with respect to devulcanization and tensile properties after revulcanization, two tread compounds with simplified formulations were prepared: one based on carbon black as filler, as reference; and another one based on silica. See Table 8, with formulation 1 for carbon black and 2 for the silica-based compound. They were mixed using the 390 mL Brabender tangential mixer according to the procedures in Table 9 for the carbon black compound and Table 6 for the silica-based one. The compounds were vulcanized into sheets of 200 mm × 200 mm × 4 mm for t${}_{90}$+4 min at 170 ${\;}^{\circ }C$ because of the thickness of the mould.

A part of both vulcanizates were ground at room temperature, sieved with the laboratory set of sieves to obtain a fraction of 2–3.5 mm and subsequently devulcanized. Devulcanization of these small batches was performed in the Brabender mixer with a chamber volume of 50 mL. The mixer was preheated to a temperature of 220 ${\;}^{\circ }C$, filled with a premix of the material to be devulcanized, using a fill factor of 0.6, and operated at 50 rpm for 5 min. The formulation used for the devulcanization was DBD: 3.9 wt%, TDAE: 2 wt%, TDTBP 1 wt%.

Subsequently, the devulcanizate was dumped into liquid nitrogen to cool down and to prevent contact with oxygen. Afterwards, it was post-treated on a mill in a similar way as the DGTR, as described before. The devulcanized compounds were revulcanized again, using the same formulations as were used before for the prior vulcanization with respect to the zinc-oxide content, stearic acid, sulfur, TBBS, DPG and TESPT. These components were assumed to have been fully consumed during the prior vulcanization which turned out to be a practical approximation. For preparing the revulcanizates the procedures as described in Table 4 and Table 7 were used. The compounds were revulcanized for t${}_{90}$+2 min at 170 ${\;}^{\circ }C$ under pressure in the Wickert press into 2 mm thick sheets as before.

Dumbbells were cut from all samples and tested for their stress strain properties according to ISO 37. Pictures were taken of the fracture-surfaces of the dumbbells after the tensile tests by microscopy.

Tensile properties for both compounds, prior to devulcanization, were a tensile strength of 20 MPa for the silica-based compound at 380% strain and 15 MPa for the carbon black-based at 400% strain. After de- and revulcanization, the compounds showed a tensile strength of 13 MPa and a strain of 280% for the silica-based and 13 MPa and 330% for the carbon black-based compound, as shown in Figure 10.

Hence, after revulcanization of the carbon black-based compound about 95% of the tensile strength could be recovered, but for the revulcanized silica compound this was only about 70%. Although this is significantly less compared to the carbon black-based compound, it is much better than could have been expected from the decrease in crosslink density of similar compounds as shown by the Horikx-Verbruggen diagrams in Figure 6, position 1. By comparing the fracture surfaces of the revulcanizates of both compounds, it appeared that the carbon black-based compound had a relatively smooth surface, as can be seen in Figure 11a, indication of a homogeneous devulcanization. In contrast to this, the fracture surface of the silica-based material showed a rough surface, as shown in Figure 11b.

The latter implies a lower amount of devulcanized rubber containing larger parts of non-devulcanized rubber. This must be due to the fact that the silica-silane-polymer bonds mainly consist of mono-sulfidic crosslinks which cannot be broken by the devulcanization process, as shown in Figure 8. As a consequence, the crosslinks that can be broken by the devulcanization process are only the di- and polysulfidic ones, mostly found in the network between the polymer chains. Comparing these results with those of the GTR highlights the negative influence on the structure and tensile strength due to the grinding of whole car tire rubber. It shows the improvements that can be expected when separating car tire components before processing.

## 3. Materials and Methods

### 3.1. Materials

The Ground Tire Rubber (GTR) used in this investigation was obtained from Genan, Dorsten, Germany. It is a commercial ground passenger car tire, medium grade with a normal-like size distribution of dimensions 1 to 3.5 mm, see Figure 12. It contains at least 45% rubber polymer including 10–35% NR [22]. This GTR type represents the cleanest fraction from the grinding process. By TGA the silica content was determined as 20–42 phr, depending on the sample drawn from the feedstock, with 23 phr a mean value measured on a larger sample size. All other materials are specified in Table 10.

### 3.2. Equipment

Continuous devulcanization was performed in a KrausMaffei ZE 25 UTX co-rotating twin-screw extruder (KraussMaffei Technologies GmbH, München, Germany), length 42D with D = 25 mm, with 3 de-aeration positions between the supply funnel and outlet and an elongated die, with dimensions of a rounded rectangular slit of 20 mm × 40 mm and length of 100 mm, as can be seen in Figure 5a. The screw design was based on a minimal shear concept as detailed in Figure 5b.

The extruder was operated at 10–30 rpm with a total residence time in the extruder and the elongated die of approx. 10 min at 20 rpm. To minimize oxidative degradation during the devulcanization, the extruder was equipped with nitrogen supply in the supply funnel, at the position of elements 36–37 and at the entrance to the die. The barrel temperature was set at 130 ${\;}^{\circ }C$ for the mixing section, 180 ${\;}^{\circ }C$ for the devulcanization section and 150 ${\;}^{\circ }C$ for the pressure section, see Figure 5. The devulcanizate was dropped directly onto a cooling calendar and cooled down to 40–60 ${\;}^{\circ }C$ to again prevent oxidation of the devulcanizate. The capacity of the whole setup was approx. 2 kg/h at 20 rpm. To prepare the virgin rubber compounds, a Brabender Plasticorder 350S internal mixer with a chamber volume of 390 mL was used. A Brabender Plasticorder internal mixer with a chamber volume of 50 mL was used for both small scale devulcanization and all compounding for revulcanization. A Schwabenthan laboratory mill with rolls of 200 mm length, a diameter of 80 mm and a speed ratio of 1.13 was used at 22 rpm for the final milling of the devulcanizate and for all milling after mixing. A Fritz pulverette with a mesh size of 2 mm, in combination with a set of laboratory sieves with opening sizes of 0.7 mm${}^{2}$, 2 mm${}^{2}$ and 3.5 mm${}^{2}$ was used to granulate the cured model compounds. The compounds were tested for their cure characteristics with a Rubber Process Analyzer, RPA Elite from TA Instruments, New Castle, DE19720, USA at 170 ${\;}^{\circ }C$, 0.833 Hz, and 2.89% strain according to ISO 6502. For vulcanization a Wickert WLP1600 laboratory compression molding press, WICKERT Maschinenbau GmbH, D-76829 Landau in der Pfalz, Germany was used at 170 ${\;}^{\circ }C$ and for a period of t${}_{90}$+2 min.

Modulus, tensile strength and elongation at break were determined with a Zwick tensile tester, ZwickRoell GmbH & Co, 89079 Ulm, Germany using dumbbell shaped samples according to ISO 37 type II. A Thermo Gravimetric Analysis (TGA) was performed with a TGA550 from TA Instruments, New Castle, DE19720, USA. For FTIR spectrometric analysis, a PerkinElmer Spectrum 100,PerkinElmer Inc, North Billerica, MA 01862, USA was used and SEM-EDX analysis were performed with a Jeol JSM 6400 SEM, Jeol SemAforce digital Image acquisition software and a Noran Voyager energy Dispersive X-ray analyzer, all from JEOL USA, Inc., Peabody, MA 01960, USA. For microscopy, a Keyence 3D, Keyence corporation, Itasca, IL 60143, USA was used.

## 4. Conclusions

GTR contains an increasing amount of silica over recent times, as result of the progressive use of this filler in modern tread compounds to improve tire performance. It was shown that devulcanization of such silica containing GTR is more difficult than of a carbon black-based GTR. The presence of silica in GTR has a double effect: It makes the rubber more difficult to devulcanize than GTR without silica, which leads to a coarser structure after revulcanization and to lower values of the tensile strength: a value of 3.5 MPa could only be obtained. By using additional DPG as secondary accelerator and adding new coupling agent TESPT, resp. an additional silanization mixing sequence in the revulcanization formulation, the tensile strength could be restored to 8 MPa. In Table 11, the best tensile results of the revulcanization experiments in this manuscript are summarized.

To emphasize the benefits of separating tire treads from the rest of the tires before processing, it was shown with tread compound studies with 80 phr carbon black or 90 phr silica as fillers that it must be possible to keep the tensile properties of a revulcanizate of a devulcanized silica-based compound at approximately 70%, from 20 MPa to 13 MPa, while for a carbon black-based compound at around 95%. The decrease in the tensile properties between revulcanized silica-based compounds and the original material, compared to those of the carbon black-based compounds, indicates that separation of the different tire parts before devulcanization to separate silica-containing and non-containing parts, may significantly improve the overall quality of the devulcanizates.

## Figures and Tables

**Figure 1 materials-12-00725-f001:**
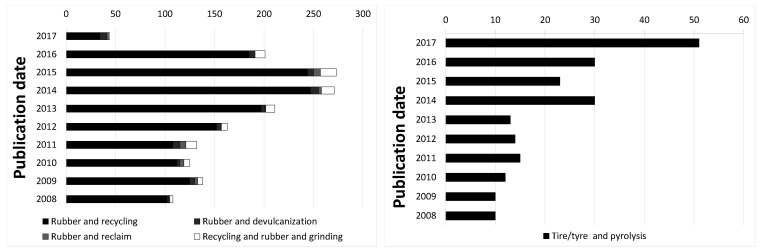
Number of patents on recycling of rubber. Countings according to Espacenet.

**Figure 2 materials-12-00725-f002:**
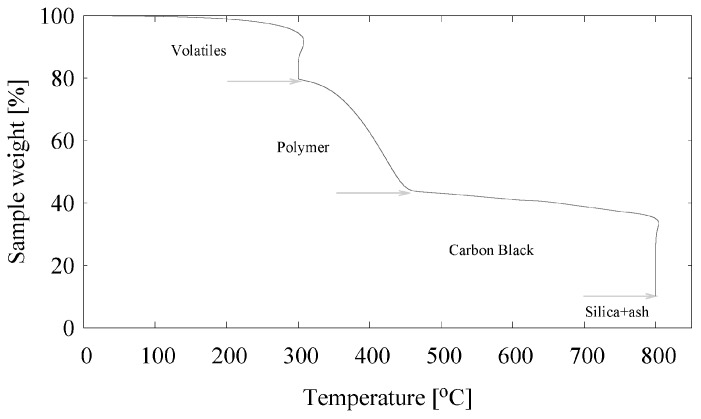
TGA analysis of the GTR.

**Figure 3 materials-12-00725-f003:**
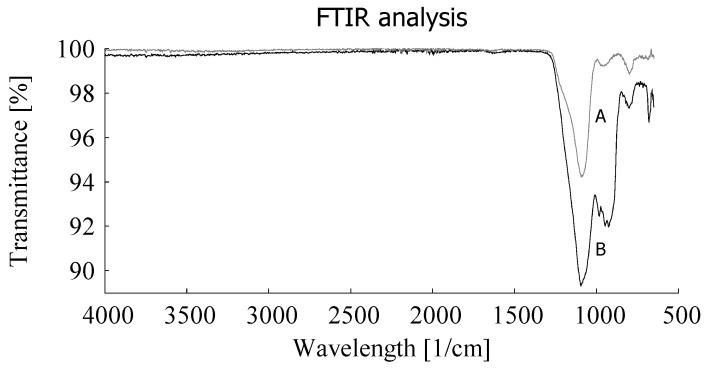
FTIR analysis of silica (**A**) as reference and ash of devulcanizate after TGA (**B**).

**Figure 4 materials-12-00725-f004:**
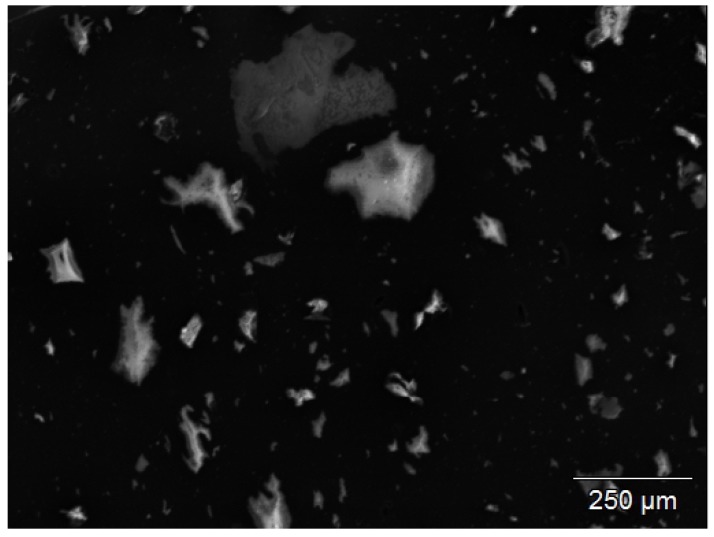
SEM-EDX picture of remaining particles from the devulcanizate after TGA. Pure silica is detected, 100–250 μm, relevant for the visual granularity of the revulcanizate.

**Figure 5 materials-12-00725-f005:**
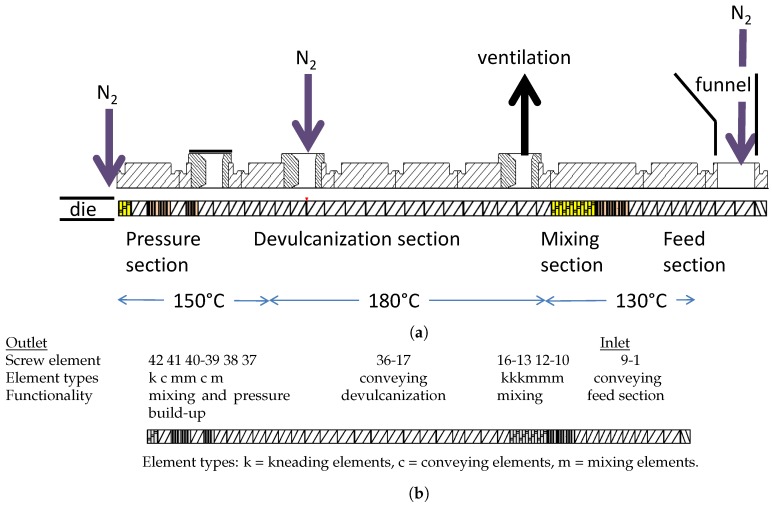
The extruder. (**a**) Layout of the extruder, from right to left. (**b**) Detailed screw design, from right to left.

**Figure 6 materials-12-00725-f006:**
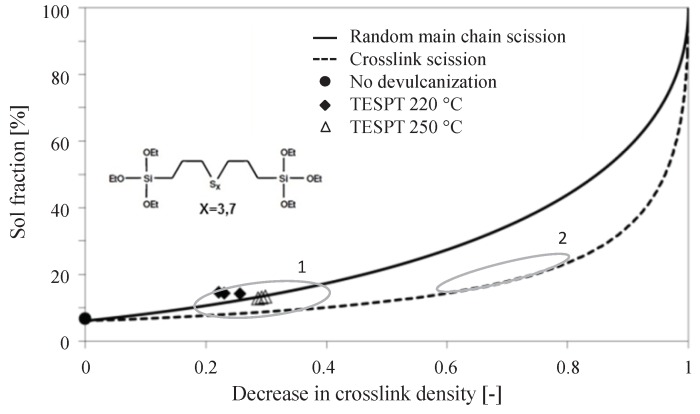
Degree of devulcanization of a silica-based tire tread compound as shown by a Horikx-Verbruggen diagram. 1 = Measurements for silica-based devulcanizates, as measured by Saiwari [14], 2 = Range of best devulcanized carbon black-based samples [5].

**Figure 7 materials-12-00725-f007:**
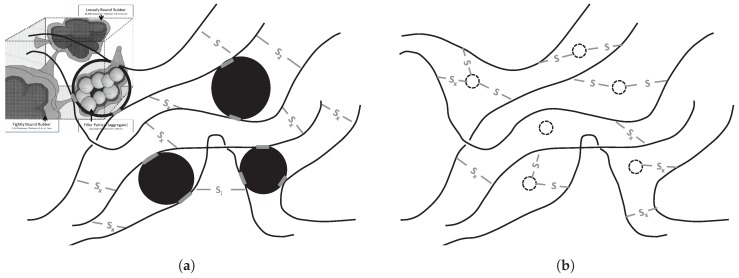
Differences in network structure between carbon black-based compounds and silica-silane-based compounds. (**a**) Sulfide bonds in carbon black rubber. Insert bound rubber by Leblanc [15]. (**b**) Sulfide bonds in silica-silane-based rubber. High amount of (mono)sulphidic bonds between silica and rubber.

**Figure 8 materials-12-00725-f008:**
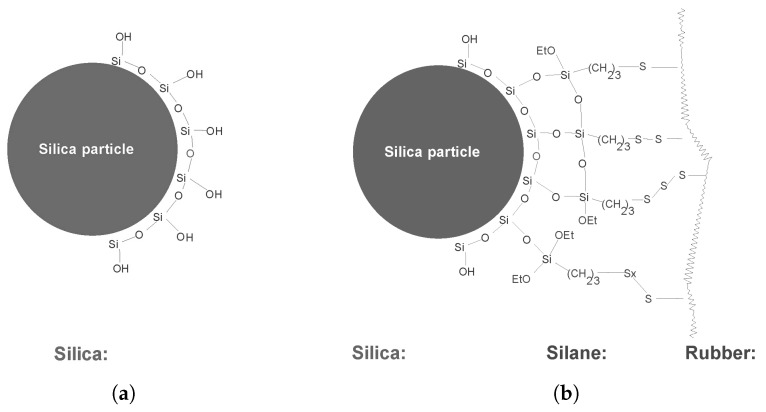
Chemical bonds between silica and rubber by silane bridges [18]. (**a**) Simplified schematics of the surface chemistry of silica. (**b**) Silane bridges between silica and rubber. Part of the silanol moieties are still unoccupied.

**Figure 9 materials-12-00725-f009:**
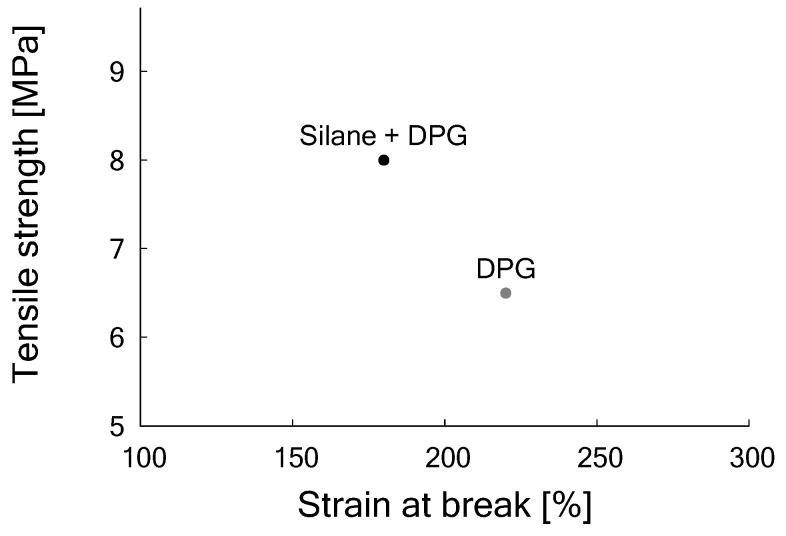
Tensile stress vs. strain at break of samples of DGTR after revulcanization with DPG only, Formulation 3 in Table 3; and with DPG and TESPT, Formulation 4 in Table 3.

**Figure 10 materials-12-00725-f010:**
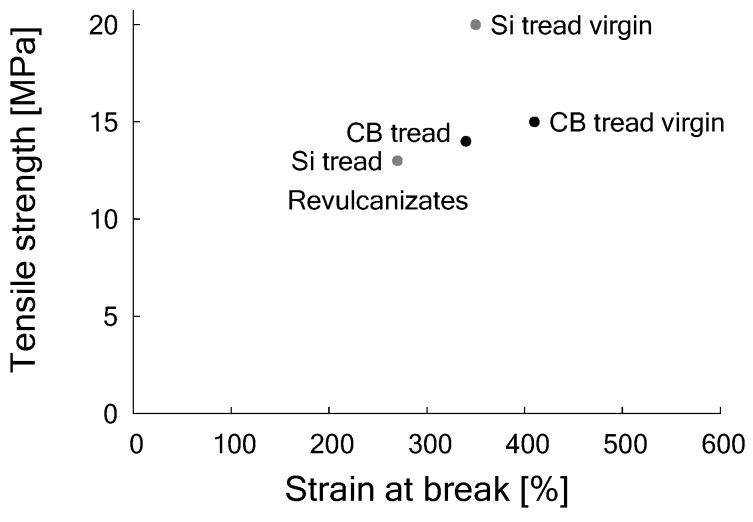
Tensile strength and strain at break of model compounds with 80 phr CB and 90 phr silica respectively, before and after devulcanization. (Re)vulcanization of each compound with the same formulation, see Table 8.

**Figure 11 materials-12-00725-f011:**
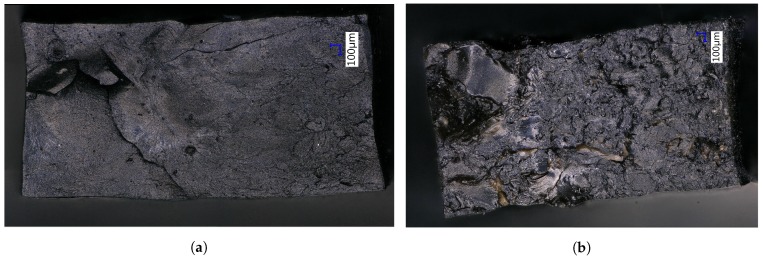
Microscopy pictures of fracture surfaces of dumbbells after tensile tests of revulcanized, carbon black-based and silica-based modelcompounds. Samples 4 mm × 2 mm. (**a**) 80 phr carbon black. (**b**) 90 phr silica.

**Figure 12 materials-12-00725-f012:**
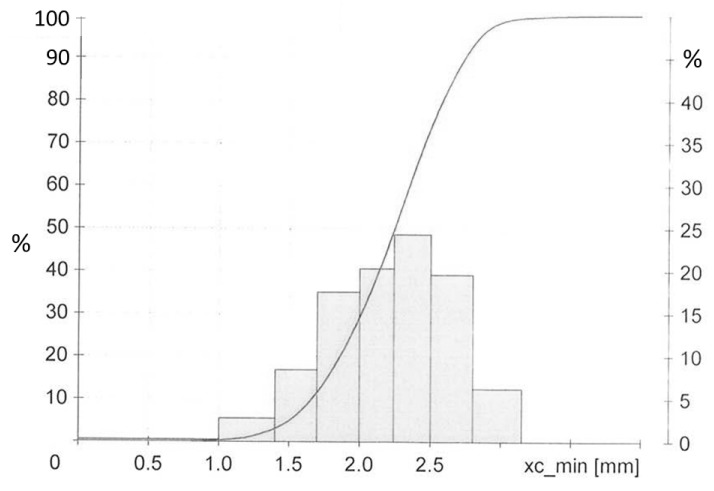
Size distribution of GTR [22].

**Table 1 materials-12-00725-t001:** TGA program for determination of the amount of silica in GTR.

Temperature	Action
	under nitrogen
⟶50 ${\;}^{\circ }C$	go to 50 ${\;}^{\circ }C$
keep at 50 ${\;}^{\circ }C$	for 2 min
50⟶300 ${\;}^{\circ }C$	100 ${\;}^{\circ }C$/min
keep at 300 ${\;}^{\circ }C$	for 15 min
300⟶620 ${\;}^{\circ }C$	20 ${\;}^{\circ }C$/min
620 ${\;}^{\circ }C$	change to air
620⟶800 ${\;}^{\circ }C$	100 ${\;}^{\circ }C$/min
keep at 800 ${\;}^{\circ }C$	for 15 min

**Table 2 materials-12-00725-t002:** Devulcanization formulations used for continuous devulcanization.

Component	Amount in wt% of GTR
DBD (2-2’-Di-Benzamido-Diphenyldisulfide)	3.9 6.85
TDAE (Treated Distillate Aromatic Extract)	0 2 5 6.2
TDTBP (Tris(2,4-Di-Tert-Butylphenyl)Phosphite)	1

All combinations of the mentioned amounts were used for continuous devulcanization.

**Table 3 materials-12-00725-t003:** Revulcanization formulations, values in phr.

Formulation nr.	1 DGTR	2 DGTR	3 DGTR	4 DGTR
Component:	re-vulca-nizate (2)	re-vulca-nizate (3)	re-vulca-nizate (4)	re-vulca-nizate (5)
Polymer (1) [x]	[100]	[100]	[100]	[100]
ZnO	3.0	2.5	2.5	2.5
Stearic acid	2.0	1.0	1.0	1.0
TDAE $\left[x\right]$	[42.7]	[42.7]	[42.7]	[42.7]
Carbon black $\left[x\right]$	[80]	[54]	[54]	[54]
Silica $\left[x\right]$		[42] (7)	[42] (7)	[42] (7)
TESPT				3.2
6PPD	1.0			
TMQ	2.0			
TBBS	1.5	1.7	1.7 + 3 (6)	1.64 + 3 (6)
DPG		2.0	2.8	2.8
Sulphur	1.5	1.4	1.4 + 3 (6)	1.64 + 3 (6)

[x] Components already present in devulcanizate, amount of DGTR is adjusted to correspond to 100 phr polymer; (1) Total polymer content of (D)GTR, a mix of mainly SBR, BR and NR; (2) Revulcanization formulation, based on the carbon black-based tread formulation; (3) DPG added because of the silica content. Amount derived from the silica-based tread formulation: (4) Amount of DPG optimized; (5) TESPT added for silanization of silica in GTR. Related to the amount of silica in the DGTR; (6) Additional sulfur and TBBS added because of the revulcanization process; (7) Based on prelimenairy experiments concerning the composition of GTR.

**Table 4 materials-12-00725-t004:** Compounding procedure for all re-vulcanizates without silane, without (*) and with DPG.

Time (min)	Processing Step
	Brabender internal mixer
	Chamber volume: 50 mL
	Fill factor: 0.6
	Initial mixer temperature of 80 ${\;}^{\circ }C$.
	Mixer set at 5 rpm rotor speed.
	Addition of devulcanizate.
0	Mixer set at 50 rpm,
1	ZnO + stearic acid,
4	TBBS + DPG (*),
4.5	Sulphur.
5	Dump.
	Homogenized for 5 min. at 0.2 mm between the rolls and sheeted off on the mill.
	Relaxation for 24 h.

**Table 5 materials-12-00725-t005:** Formulation of the silica-based tread compound as used for the analysis of the devulcanizate as shown in the Horikx-Verbruggen diagram in Figure 6.

Component	(phr)
SBR	103
BR	25
Silica Zeosil 1165	80
TESPT	7.0
TDAE	5.0
ZnO	2.5
Stearic acid	2.5
6PPD	2.0
TMQ	2.0
Sulfur	1.4
TBBS	1.7
DPG	2.0

**Table 6 materials-12-00725-t006:** Preparation procedure for silica-based tire tread formulation.

Time (min)	Processing Step
	Before silanization
	Brabender internal mixer
	Chamber volume: 390 mL
	Fill factor: 0.7
	Initial mixer temperature setting: 65 ${\;}^{\circ }C$
	Rotation speed: 100 rpm
	Total mixing time: 5 min,
	Mixing order:
0	Polymers,
1	33% (silica, silane and TDAE).
2	33% (silica, silane and TDAE).
3	remaining (silica, silane and TDAE).
4	ZnO and stearic acid,
5	6PPD + TMQ,
	Silanization.
6	Continue with speed at 100 rpm until 145 ${\;}^{\circ }C$ has been reached
6	Lower the speed to approx. 80 rpm to keep the temperature at 145 ${\;}^{\circ }C$
	Dump after 0.5 min mixing at 145 ${\;}^{\circ }C$.
	Cooled down and sheeted off on the mill.
	Relaxation for 24 h.
	Addition of curatives.
	Initial mixer temperature: 50 ${\;}^{\circ }C$ at 75 rpm.
	Mixing order:
0	Silanized compound,
1	TBBS + DPG + sulfur,
max 3	Dump at 100 ${\;}^{\circ }C$
	Cooled down and sheeted off on the mill to 2 mm thick slabs.
	Rest for 12 h.

**Table 7 materials-12-00725-t007:** Compounding procedure for re-vulcanizate formulations with silane (TESPT).

Time (min)	Processing Step
	Brabender internal mixer
	Chamber volume: 50 mL
	Fill factor 0.6
	The mixer temperature was set to 145 ${\;}^{\circ }C$.
	The devulcanizate was added at 5 rpm rotor speed.
	Silanization.
0	Mixer set at 50 rpm rotor speed.
1	Silane.
5	Dump at approx. 145 ${\;}^{\circ }C$.
0	The devulcanizate was cooled down to 60 ${\;}^{\circ }C$ and milled with a gap-width of 0.1–0.5 mm.
5	Sheeted off at 2 mm.
	Relaxation for 72 h.
	Addition of curatives.
	Initial mixer temperature of 50 ${\;}^{\circ }C$ and 50 rpm rotor speed.
	Mixing order:
0	Silanized devulcanizate,
0.5	ZnO + stearic acid,
1	TBBS + DPG,
1.5	Sulphur.
2	Dump.
	Homogenized for 5 min. at 0.2 mm between the rolls and sheeted off on the mill.
	Relaxation for 24 h.

**Table 8 materials-12-00725-t008:** Vulcanization formulations of tread compounds, values in phr.

Recipe nr. Component:	5 CB Based Tread Compound	6 Silica-Based Tread Compound (1)
SBR	65	70
BR	35	30
ZnO	3.0 [*]	2.5 [*]
Stearic acid	2.0 [*]	1.0 [*]
TDAE	35	32.5
Carbon Black N550	80	
Silica Zeosil 1165MP		90
TESPT		7.2 [*]
6PPD	1.0	2.0
TMQ	2.0	2.0
TBBS	1.5 [*]	1.7 [*]
DPG		2.0 [*]
Sulphur	1.5 [*]	1.4 [*]

(1) Based on the Green Tire Michelin patent [12], $\left[*\right]$ Components used for revulcanization.

**Table 9 materials-12-00725-t009:** Compounding procedure for carbon black-based tire tread formulation.

Time (min)	Processing Step
	Brabender internal mixer
	Chamber volume: 390 mL
	Fill factor: 0.7
	Mixer temperature setting: 50 ${\;}^{\circ }C$
	Rotation speed: 50 ${\;}^{\circ }C$ at 50 rpm
	Mixing order:
0	Polymers,
1	ZnO + stearic acid,
2	33% (CB + TDAE),
3	33% (CB + TDAE),
4	remaining (CB + TDAE),
5	6PPD and TMQ,
6	TBBS,
6.5	Sulfur.
7	Dump temperature max. approx. 80 ${\;}^{\circ }C$.
	Sheeted off on a mill to 2 mm thick slabs.
	Rest for 12 h.

**Table 10 materials-12-00725-t010:** Materials employed.

	Designation	Supplier
	Devulcanization agents:	
DBD	2-2’-DiBenzamidoDiphenyldisulfide	Schill and Seilacher GmbH, Boeblingen, Germany
TDTBP	Tris(2,4-Di-Tert-Butylphenyl)Phosphite	Sigma Aldrich Cooperation, Zwijndrecht, The Netherlands
	Processing oil:	
TDAE	Treated Distillate Aromatic Extract, VIVATEC 500	Hansen & Rosenthal, Hamburg, Germany
	Polymers and fillers:	
SBR	Styrene Butadiene Rubber grade SPRINTAN SLR 4601 (50% vinyl, 25% styrene)	Trinseo Deutschland GmbH, Schkopau, Germany
BR	Butadiene Rubber grade BUNA CB24	Arlanxeo Deutschland GmbH, Leverkusen, Germany
CB	Carbon Black grade N550	Evonik Carbon Black GmbH, Essen, Germany
Silica	Silica type Zeosil 1165MP	Rhodia Silices, France
	Vulcanization system:	
S	Sulfur	Industry standard
TBBS	N-Tert-Butyl-2-BenzothiazoleSulfenamide	Industry standard
ZnO	Zinc Oxide	Industry standard
St.A.	Stearic acid	Industry standard
DPG	1,3-DiPhenylGuanidine	Industry standard
	Stabilizers system:	
6PPD	N-(1,3-dimethylbutyl)-N’-Phenyl-p-PhenyleneDiamine	Industry standard
TMQ	2,2,4-Trimethyl-1,2-DihydroQuinoline	Industry standard
	Coupling agent:	
TESPT	bis[3-(TriEthoxySilyl)Propyl] Tetrasulfide	Evonik Industries AG, Essen, Germany

**Table 11 materials-12-00725-t011:** Summary of tensile properties of (re)vulcanizates, $\left[*\right]$ Components for revulcanization only.

Compound	Revulcanization Table 3 and Table 8	Tensile Strength Formulation, (MPa)	Strain @ Break (%)
DGTR	1	3.5	250
,,	2	5.0	180–230
,,	3	6.5	180–220
,,	4	8.0	140–170
CB compound	5	15	400
Devulcanized CB compound	5 [*]	14	330
Silica compound	6	20	360
Devulcanized silica compound	6 [*]	13	250

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
