# Peer review of "Implications of the Use of Silica as Active Filler in Passenger Car Tire Compounds on Their Recycling Options"

_materials, 2019, doi:10.3390/ma12050725_

Reviewer 1 Report

Dear Authors,

This paper is very interesting and could be considered for publication. However, before the final decission the major revision should be performed.

My detailed comments in the attachment.

Author Response

Please find our responses to your remarks in the pdf file

Reviewer 2 Report

The increasing usage of green tire caused a sharp increase of tire rubber reinforced with silica, understanding the devulcanization of this kind of tire rubber is very important. The paper had some new findings and it is interesting. However, several points need to be further improved.

  P 3 Line 84, polymer content is 35% is unacceptable, pls check,

Fig. 10 losing the label of X-axis

Fig. 11 put the scale in the picture

Fig. 3 is not in standard form

P13 L246, can you give the size distribution?

Most of tables are suggested to put in Sec. 3 Materials and Methods

L127 what does +_ mean?

Table 4 Thoroughly homogenized , what is the degree? Is there any scale?

Author Response

Please find our comments on your remarks in the word document.

Reviewer 3 Report

•  Have the authors got any reference that use the stress-strain properties to analyze the quality of devulcanization process?. Therefore, authors need to improve the explanation why their use this method of analysis.

•  Several time the authors use the sentence “Vulcanization was performed under pressure in the Wickert press for t90+2 minutes at 170”. But, is the optimum cure time the same for all the revulcanization samples?, and what long is each vulcanization?

•  Table 3 is confused and needs more and better explanation. Is "polymer" GTR with all components (SBR, NR, BR, CB, Silica, S and other..)?. It means that formulation 1 incorporate and average of 23phr of silica? And samples 2,3,4  an additional amount of 42phr?

•  Figure 6 compare silica based devulcanized and CB based devulcanized, are both from Saiwari (21) study or only the silica samples?.  This point also needs better explanation.

• Figure 9. There are an error in the caption. I think that is table 3 not table 8.

·      In the additional study of point 2.5 “Benefits of using pure……”  It would be necessary a study using silica based compound without silane agent TESPT.            

Author Response

Please find our comments on your remarks in the word document.

Round  2

Reviewer 3 Report

With the improvements made in the manuscript and the answere to the comments I accept in present form